# AlphaFold Accurately Predicts the Structure of Ribosomally Synthesized and Post-Translationally Modified Peptide Biosynthetic Enzymes

**DOI:** 10.3390/biom13081243

**Published:** 2023-08-12

**Authors:** Catriona H. Gordon, Emily Hendrix, Yi He, Mark C. Walker

**Affiliations:** Department of Chemistry and Chemical Biology, University of New Mexico, Albuquerque, NM 87131, USA

**Keywords:** RiPPs, AlphaFold, graspetide

## Abstract

Ribosomally synthesized and post-translationally modified peptides (RiPPs) are a growing class of natural products biosynthesized from a genetically encoded precursor peptide. The enzymes that install the post-translational modifications on these peptides have the potential to be useful catalysts in the production of natural-product-like compounds and can install non-proteogenic amino acids in peptides and proteins. However, engineering these enzymes has been somewhat limited, due in part to limited structural information on enzymes in the same families that nonetheless exhibit different substrate selectivities. Despite AlphaFold2’s superior performance in single-chain protein structure prediction, its multimer version lacks accuracy and requires high-end GPUs, which are not typically available to most research groups. Additionally, the default parameters of AlphaFold2 may not be optimal for predicting complex structures like RiPP biosynthetic enzymes, due to their dynamic binding and substrate-modifying mechanisms. This study assessed the efficacy of the structure prediction program ColabFold (a variant of AlphaFold2) in modeling RiPP biosynthetic enzymes in both monomeric and dimeric forms. After extensive benchmarking, it was found that there were no statistically significant differences in the accuracy of the predicted structures, regardless of the various possible prediction parameters that were examined, and that with the default parameters, ColabFold was able to produce accurate models. We then generated additional structural predictions for select RiPP biosynthetic enzymes from multiple protein families and biosynthetic pathways. Our findings can serve as a reference for future enzyme engineering complemented by AlphaFold-related tools.

## 1. Introduction

It has been estimated that approximately one third of small-molecule drugs are chemicals produced by living organisms or are derived from those compounds [1]. Due to the success of these compounds, or natural products, there has long been interest in using the enzymes that produce them to generate new compounds that do not exist, or have not yet been found, in nature. The biosynthesis of one class of natural products, ribosomally synthesized and post-translationally modified peptides (RiPPs), seems to be particularly amenable to this approach. RiPPs are a large and growing class of natural products with biological activities ranging from antibiotic to antiviral [2]. RiPPs are biosynthesized from a genetically encoded precursor peptide that is extensively post-translationally modified through the installation of macrocycles, heterocyclization of the amide backbone, and the epimerization of amino acid residues, among other modifications [3]. This precursor peptide contains a region (a leader peptide at the N-terminus, a follower peptide at the C-terminus, or both) that is recognized and bound by the enzymes that post-translationally modify the peptide, and a region called the core peptide where these post-translational modifications are installed. Many of these biosynthetic enzymes are able to install the same post-translational modification in multiple locations on the core peptide using the same active site, thus being able to act on structurally distinct substrates (Figure 1) [3,4]. At the same time, these enzymes are highly accurate, producing a single product from up to thousands of chemically possible molecules. Due to this built-in substrate tolerance, RiPP biosynthetic enzymes have generated a great deal of interest as catalysts to generate libraries of natural-product-like compounds that can be screened for new biological activities [5,6,7,8].

However, efforts to rationally engineer these enzymes to increase their substrate scope even further or allow them to install post-translational modifications at altered locations in the core peptide have been limited. These engineering efforts would be supported by having access to a large amount of structural information about these enzymes—the acquisition of which, to date, has been hampered by the relatively low throughput of experimentally obtaining it. 

The field of protein structure prediction has garnered significant attention over the years, largely due to its potential to revolutionize our understanding of biological processes and facilitate drug discovery. Physics-based modeling is one avenue that scientists have explored extensively, employing a series of robust algorithms and methodologies to predict the three-dimensional conformation of protein chains. Popular tools under this category include Chemistry at Harvard Molecular Mechanics (CHARMM) [9] and Assisted Model Building with Energy Refinement (AMBER) [10]. To reduce the computational costs, physics-based coarse-grained force fields, such as the United-RESidue (UNRES ) model [11,12,13], which simplifies the protein structure for efficiency, were also quite popular in early days. Parallel to these physics-based endeavors, the closing years of the 20th century and the onset of the 21st century experienced a surge in the popularity of homology modeling. This technique leverages the evolutionary linkages between proteins to predict their structures. Essentially, proteins that have evolved from a common ancestor, termed homologs, are presumed to maintain structural congruencies. This concept was highlighted by researchers like Sander and Schneider in their groundbreaking 1991 paper, which asserted that the structures of proteins can often be predicted with high accuracy using homologous proteins as templates [14]. This notion not only simplified the complex puzzle of protein folding but also bridged the gap between evolutionary biology and structural bioinformatics, emphasizing that the history embedded in protein sequences can provide valuable insights into their three-dimensional architectures. 

The release of AlphaFold has significantly improved prediction accuracy and opened the possibility of rapidly obtaining computationally predicted structural information [15]. Following the launch of AlphaFold2, a multimeric version of AlphaFold2 was introduced [16], designed specifically to predict the structures of protein complexes. Owing to its superior accuracy, AlphaFold 2 and the AlphaFold 2 multimer version have been employed in the study of intrinsically disordered proteins, complex biomolecular systems, and other structurally challenging systems [17,18]. Some independent evaluations of AlphaFold-predicted monomer structures for drug development purposes, and the accuracy of predicted loop regions, suggest that AlphaFold has achieved commendably high precision in terms of structure prediction [19,20]. On the other hand, evaluations of the AlphaFold multimer version using independent complex datasets have led to mixed conclusions [21,22,23,24]. More importantly, while only minimal inputs are required to run AlphaFold, certain parameters can significantly influence the accuracy of AlphaFold’s predictions. Given that RiPP biosynthetic enzymes represent a group of under-studied protein complexes, which can undergo conformational changes upon precursor peptide binding, it is both urgent and necessary to assess and parameterize the effectiveness of this deep learning program in predicting the three-dimensional structures of RiPP biosynthetic enzymes.

## 2. Materials and Methods

### 2.1. Modeling ATP-Grasp Ligase Family Enzymes

A set of seven different enzymes belonging to the ATP-grasp ligase family were modeled using the ColabFold v1.3 implementation of the AlphaFold 2.1 with the mmseq2 software package (installed locally) [15,16,25,26] both as monomers and dimers. The different enzymes utilized for this portion of the study included the RiPP biosynthetic enzymes MdnC (5IG9) [27], MdnB (5IG8) [27], aMdnB (7M4S) [28], CdnC (7MGV) [29], and PsnB (7DRM) [30], as well as ATP-grasp ligases that are not involved in RiPP biosynthesis, ArgX (3VPB) [31] and LysX (3VPD) [31]. The two enzymes that are not involved in RiPP biosynthesis were chosen due to their structural similarity to the RiPP biosynthetic enzymes, as well as their thorough structural and mechanistic characterization.

Various sets of parameters were tested, including different combinations of recycle numbers, the use of templates, and the utilization of AMBER relaxation refinement. AlphaFold produces five models for each prediction run, and subsequently ranks them from 1 to 5; these rankings are determined based on the predicted Local Distance Difference Test score (pLDDT). The recycle number indicates the quantity of iterative refinements that predictions undergo over the course of the run. The use of a template instructs AlphaFold to search the pdb70 database for the top 20 templates containing the highest number of residues correctly aligned to the input sequence; the network additionally offers ‘bad’ templates to ensure that the program does not directly copy the templates. The final parameter involves Assisted Model Building with Energy Refinement (AMBER), which relaxes the models using a restrained energy minimization process to preserve stereochemical plausibility.

### 2.2. Set Parameters

The seven aforementioned enzymes were modeled as monomers, in conjunction with model type AlphaFold2-ptm and recycle possibilities of 3, 12, 24, 48, and 72. Note that no AMBER refinements or templates were used for this first set. The second and third sets employed the same seven enzymes, model type, and possible recycle numbers (i.e., 3, 12, 24, 48, and 72), but used templates with no AMBER refinement, and templates with AMBER refinement, respectively. Three sets of runs with the same parameters were conducted for the prediction of dimer complexes, though notably, the model type was changed to AlphaFold2-MultimerV1. MdnB ATP Grasp Ligase (5IG8) was excluded from the dimer models’ assessment, as discrepancies in the available PDB file prevented proper analysis.

### 2.3. Application of Pipeline to Other RiPP Biosynthetic Enzymes

Based on the results of parameterization, it was determined that the predictive models that used 48 recycles and pdb70 templates and omitted AMBER refinement would be used. Subsequently, these parameters were applied to RiPP biosynthetic enzymes outside of the ATP grasp ligase family, for the purpose of gauging the use of AlphaFold with these criteria on a wider scale. Representative enzymes were chosen from several different RiPP families. The set included NisB (PDB ID: 4WD9) [32], CylM (5DZT) [33], PtbD (5W99) [34], TbtD (5WA4) [34], TbtB (6EC8) [35], TruD (4BS9) [36], YcaO (6PEU) [37], PatA (4H6V) [38], Oph-DC6 (5N0Q) [39], PCY1 (5UW3) [40], PaaA (5FF5) [41], Lasso Peptide Synthetase B1 (6JX3) [42], MccB (6OM4) [43], BamL (4KVZ) [44], BpumL (4KWC) [44], CypD (6JDD) [45], PagF (5TTY) [46], NosA (4ZA1) [47], and DurN (6C0Y) [48]. These enzymes were modeled according to their reported biological oligomerization (e.g., monomer or dimer), in order to better inform future wet-lab studies. The results from this portion of the study were compared to their relevant PDB crystal structures; monomers were compared to all monomers in the PDB file, while dimers were compared to all biologically relevant dimers in the PDB file, if more than a single dimer was present. The US-align program [49] and Bio3D [50] were used to yield the TMscore and RMSD values for analysis.

### 2.4. Evaluation of Models

To properly evaluate the accuracy of the AlphaFold predictions, the root mean square deviation (RMSD) and TMscore values were calculated for each predictive model relative to their corresponding experimental structures. RMSD is typically used to assay the distances between atoms of the predicted structure and the reported PDB structure. However, the use of RMSD as an assessment factor poses the risk of producing biased analysis; RMSD can potentially be deceptively high or low, depending upon the alignment coverage and sequence length [51]. The TMscore, however, was developed to circumvent such biases, through the use of a protein size-dependent scale and by assessing all residue pairs [51]. Consequently, both assessment factors were utilized to gain a rounded understanding of the accuracy of the predictive models. The ATP grasp ligase predictions that correspond to 48 recycles (both with and without templates/AMBER) were assessed using US-align [49] to produce RMSD values. 

## 3. Results

The AlphaFold-predicted monomer structures were compared to all the monomers in their cognate experimental structures. The predicted structures were similar to the experimental structures (Figure 2), with TMscores ranging from 0.8794 to 0.9979, and RMSD values ranging from 0.28 Å to 4.31 Å (Figure 3). Even with the lowest number of recycles, no use of templates, and no AMBER refinement, the results were remarkably accurate, with TMscores ranging from 0.8808 to 0.9823 and RMSD values ranging from 0.79 Å to 4.31 Å. These values are ideal, as a TMscore of 1 is considered perfectly aligned, and values over 0.5 suggest a roughly accurate alignment [51], and the generally accepted RMSD value for describing accurately predicted structures is ≤1.50 Å. The predicted structures of the enzymes not involved in RiPP biosynthesis were more consistently similar to their experimental structures, with TMscores ranging from 0.9796 to 0.9979, and RMSD values ranging from 0.28 Å to 0.94 Å (Appendix A). The predicted structures of RiPP biosynthetic enzymes exhibited a greater range of similarity to their experimental structures, with TMscores ranging from 0.8794 to 0.9965 and RMSD values ranging from 0.42 Å to 4.31 Å. 

To gain context for these ranges, we compared every monomer in an experimental structure to every other available monomer in that same structure and found that the TMscores ranged from 0.7779 to 0.9998, while the RMSD values ranged from 0.09 Å to 4.24 Å. Again, the monomers from enzymes not involved in RiPP biosynthesis are more consistently similar to each other than those from enzymes involved in RiPP biosynthesis (Appendix A). This intra-experimental structure comparison revealed that AlphaFold-predicted monomers are within the same range of similarity to experimental structures as monomers from experimental structures are to each other. These differences between experimental structures are responsible for the apparent multimodal distribution of TMscores and RMSD values (Figure 3). When the monomers in an experimental structure adopt different conformations, the predicted structures are more similar to one monomer than the other, resulting in a set of values that show high similarity and a set of values that show similarity like that of the comparison between the experimental monomers. The larger spread in the similarity for RiPP biosynthetic enzymes could be due to a number of experimental structures including bound precursor peptide, the binding of which is known to cause structural rearrangements [27,28,29,30].

Overall, AlphaFold performed similarly regardless of the different parameters on the initial set of monomer ATP grasp ligase predictions (Figure 3, Appendix A). A modest improvement in the mean TMscore and RMSD values was observed upon the usage of templates, with a further modest improvement in TMscore at 48 recycles. However, these differences were not statistically significant. The improvement in the mean scores between 24 recycles with templates and 48 recycles with templates was largely due to improved predictions for two proteins: LysX and CdnC. Between 24 recycles with templates and 48 recycles with templates, the RMSD values for the most similar models improved from 0.53 Å to 0.30 Å for LysX and 0.89 Å to 0.42 Å for CdnC, with similar concomitant improvements in TMscore. Notably, there were no improvements in the TMscore or RMSD values in the predictions produced with both templates and AMBER versus those produced solely with templates (i.e., without AMBER refinement). For example, including the use of templates for models predicted with 48 recycles increased the TMscore for 99 (76%) alignments while decreasing the TMscore for 23 (18%) alignments, but when AMBER relaxation was added, the TMscore increased for 15 (12%) alignments and decreased for 100 (77%) alignments (Appendix A). Similar patterns for improved and worsened alignments were observed in RMSD, as well. The changes that occurred upon including AMBER refinement were, however, relatively small in magnitude.

Similar patterns were observed for the dimer predictions (Figure 2). It was found that 72 recycles were unnecessary due to the lack of apparent RMSD value improvement and the significant additional computational cost (Figure 3). Thus, it was determined that the use of 48 recycles with templates would be sufficient to ensure the production of accurate structural predictions, without the computational cost of using a higher recycle number or the potential pitfalls of AMBER.

When experimental structures for the enzymes of interest are available, it is possible to determine which of the five models produced by AlphaFold is most similar to the experimental structure. However, in the absence of an experimental structure, the ranking system of AlphaFold is all the information present. In order to evaluate the predictive ranking system of AlphaFold, the average RMSD was calculated for ranks 1–5 across the set of ATP grasp ligase monomers and dimers. Though there are no overall trends regarding the direct correlation between predicted rank and RMSD, this analysis provided further support that there seems to be no significant difference between models produced with templates versus models produced with templates and AMBER refinement. A further comparison of the best model produced from each ATP grasp ligase enzyme (with the various parameters tested) and its corresponding rank 1 model revealed that AlphaFold’s predicted ranking system does not always reflect the most accurate (or ‘best’) model (Figure 4). Moreover, this is true for both the monomer and dimer prediction sets. However, aligning the predicted monomer structures to each other revealed they were broadly more similar to each other than the experimental structures were to each other (Appendix A), suggesting that the use of the rank 1 model would give similar results to those obtained using the most similar model, were experimental data available.

To explore the applicability of AlphaFold for RiPP biosynthetic enzymes that are not members of the ATP-grasp ligase family, predicted structures were generated for select enzymes with experimental structures available (Appendix A). Across the wider set of RiPP enzyme models (excluding the preliminary ATP Grasp Ligase-predicted structures), the average TMscore was 0.9727, with maximum and minimum values of 0.9980 and 0.8062, respectively (Figure 5). The range of RMSD values for this set of predictions had extrema of 0.23 Å and 5.26 Å. Some predictions, such as the NisB models, displayed high TMscores but relatively large RMSD values—which was likely due to their slightly lower sequence coverage (0.944 versus 1.000 for other enzymes). However, the majority of the models followed the expected correlation of low RMSD coupled with a high TMscore. The previously stated most similar TMscore and RMSD of 0.9980 and 0.23 Å correspond to CypD (6JDD) Model 1 (rank 3); the least similar TMscore and RMSD values (0.8062 and 5.26 Å, respectively) belong to YcaO (6PEU) Model 4 (rank 5) compared to the dimer of chains C and D from the experimental structure. Despite the significantly larger RMSD value for YcaO Model 4-C/D, its TMscore was still relatively close to 1.0, indicating that the model is still a sufficiently accurate prediction of the reported PDB structure. However, given that the other YcaO models exhibited higher TMscore and RMSD values, it is advised to use the highest scoring predictions for further studies. Furthermore, our examination of the TMscores and RMSD values with respect to ranking implied that models ranked first and second may, on average, produce higher TMscores—but it should be emphasized that is not the case for all predictive models.

CylM and NisB are well-characterized RiPP biosynthetic enzymes, and thus, were chosen as a focus for this study (Figure 6). Both enzymes are lanthipeptide synthases responsible for carrying out post-translational modifications (specifically dehydration and subsequent cyclization reactions for CylM [33] and dehydration for NisB [32]) on their respective peptide substrates. The five predictive models for CylM (5DZT) yielded an average TMscore of 0.9530 and an average RMSD of 2.11 Å. The ‘best’ prediction for CylM was provided by Model 1 (rank 4), with TMscore and RMSD values of 0.9954 and 0.72 Å. These values convey a highly accurate predictive model. There was little variance amongst the TMscores for all five CylM models, but there was a sharp drop-off in the RMSD value in models 3–5, which indicates greater deviation between the sub-structures of models 1–2 and models 3–5. Moreover, there was an apparent correlation between RMSD and TMscore for all NisB models, as exemplified by the linear inverse trends in Figure 6. The NisB models displayed higher TMscores overall (relative to CylM), with an average value of 0.9924, and with a maximum TMscore of 0.9928. However, it should be noted that despite the increased TMscores, the average RMSD for the NisB models was 3.94 Å; this would otherwise suggest less accurate structures, if not for the previously listed limitations of RMSD as an assessment tool.

## 4. Discussion

The revelations drawn from this study illuminate the potential of computational tools in the rapidly advancing field of RiPP biosynthetic enzyme study. AlphaFold’s adeptness in predicting structures using ColabFold’s default parameters is pivotal, underscoring the software’s robust algorithmic foundation. The fact that certain protein structures are discernibly better predicted with an augmented cycle count, specifically 48 recycles, coupled with the use of templates, emphasizes the importance of parameter optimization. This could allude to a more intricate folding mechanism or a higher degree of secondary structural elements in those proteins that benefit from such an increase. Further research might delve deeper into identifying any sequences or structural motifs within these proteins that contribute to this variation in prediction accuracy. 

Furthermore, the capacity of AlphaFold to consistently generate five viable models for almost every RiPP enzyme sequence evaluated is noteworthy. This not only affirms the reliability and repeatability of the software but also suggests its broad-spectrum applicability. The implications of this are manifold. For one, this could drastically reduce the time and resources conventionally spent on elucidating enzyme structures through experimental methods such as X-ray crystallography or NMR spectroscopy. More significantly, the potential to understand structure–function relationships at a higher throughput presents an unprecedented opportunity. The structural configurations of enzymes, especially those involved in biosynthetic pathways, are inherently linked to their catalytic roles. By having ready access to accurate models, the intricate mechanistic pathways can be explored with higher precision. This might offer insights into the nuances of substrate recognition, catalytic transition states, or even allosteric regulation sites, which traditionally remain elusive. 

This study has far-reaching implications for RiPP engineering, an avenue that has gained momentum with the increasing demand for novel natural products or their analogs with desirable biological activities. By having a foundational knowledge of the enzyme’s 3D architecture, it is much easier to design mutations, domain swaps, or entirely synthetic enzymes to synthesize novel compounds, potentially leading to groundbreaking therapeutics or other industrially relevant products. While the present study underscores AlphaFold’s efficacy in predicting RiPP enzyme structures, it also sets the stage for an era where computational tools and experimental biology seamlessly intertwine, facilitating a more comprehensive understanding and manipulation of nature’s molecular machinery.

## Figures and Tables

**Figure 1 biomolecules-13-01243-f001:**
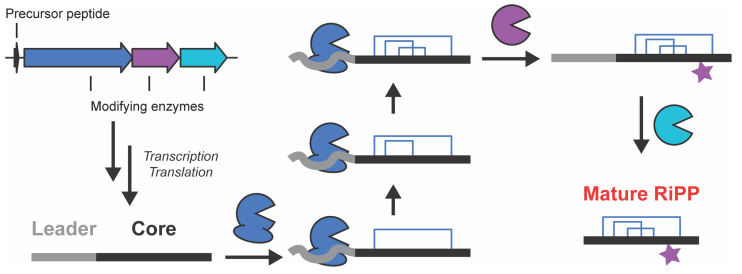
RiPP Biosynthesis. Schematic representation of the biosynthetic logic of RiPP production.

**Figure 2 biomolecules-13-01243-f002:**
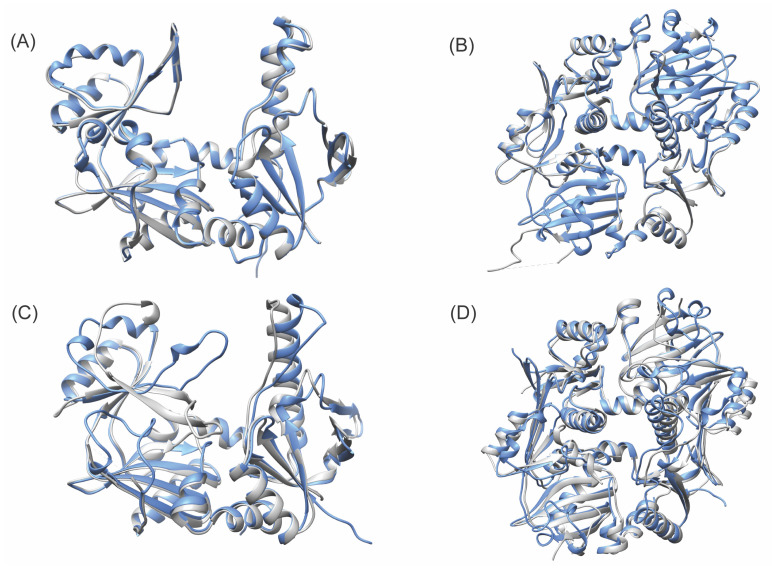
Comparison of experimental structures and AlphaFold-produced models. (**A**,**B**): RiPP biosynthetic ATP-grasp ligases with the highest-TM-score AlphaFold models (blue) compared to experimental structures (gray). (**C**,**D**): RiPP biosynthetic ATP-grasp ligases with the lowest-TM-score AlphaFold models compared to experimental structures. (**A**) CdnC monomer (7MGV, Chain B), 48 recycles with templates, rank 2. (**B**) MdnC dimer (5IG9, Chains C and D), 3 recycles, template, rank 2. (**C**) MdnB monomer (5IG8, Chain A), 24 recycles, template, AMBER, rank 3. (**D**) aMdnB dimer (7M4S, Chains B and C), 48 recycles, template, rank 3.

**Figure 3 biomolecules-13-01243-f003:**
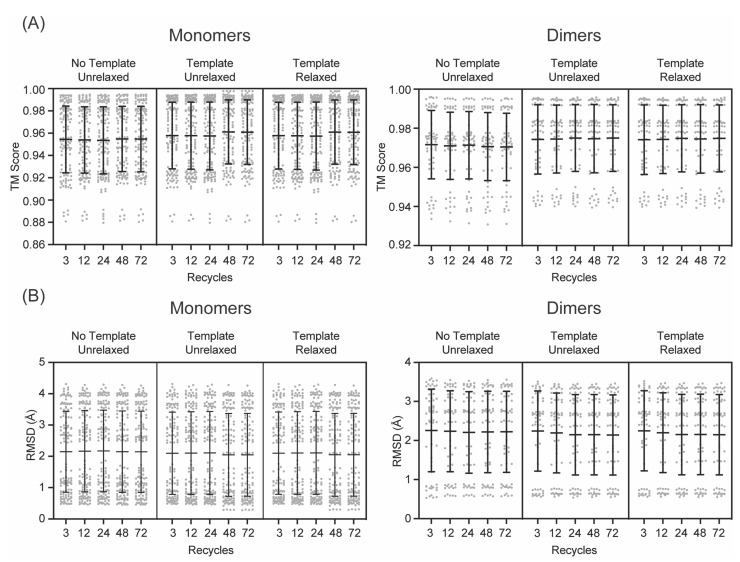
Accuracy of AlphaFold models versus experimental structures of ATP-grasp ligase family enzymes. All AlphaFold model monomers and dimers were compared to all experimental monomers and all biologically relevant experimental dimers from their cognate experimental structures. Mean TMscores (**A**) and RMSD (**B**) are represented by horizontal lines. Error bars represent standard deviations, and gray circles are values from individual comparisons.

**Figure 4 biomolecules-13-01243-f004:**
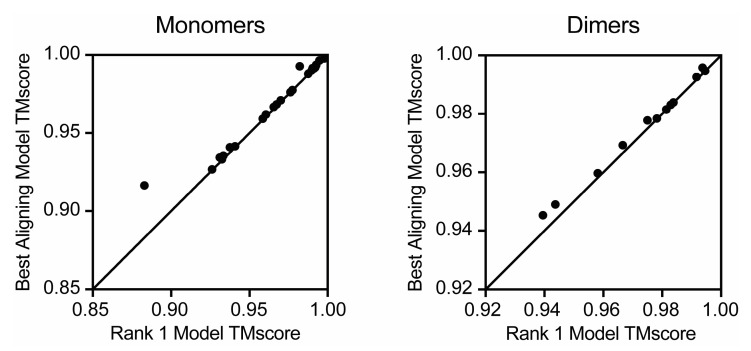
Comparison of TMscore values of highest scoring and rank 1 models. TMscores from the rank 1 AlphaFold monomers and dimers produced with 48 recycles and templates compared to each monomer or biologically relevant dimer from their cognate experimental structures. The y = x line highlights equality and is not a line of best fit.

**Figure 5 biomolecules-13-01243-f005:**
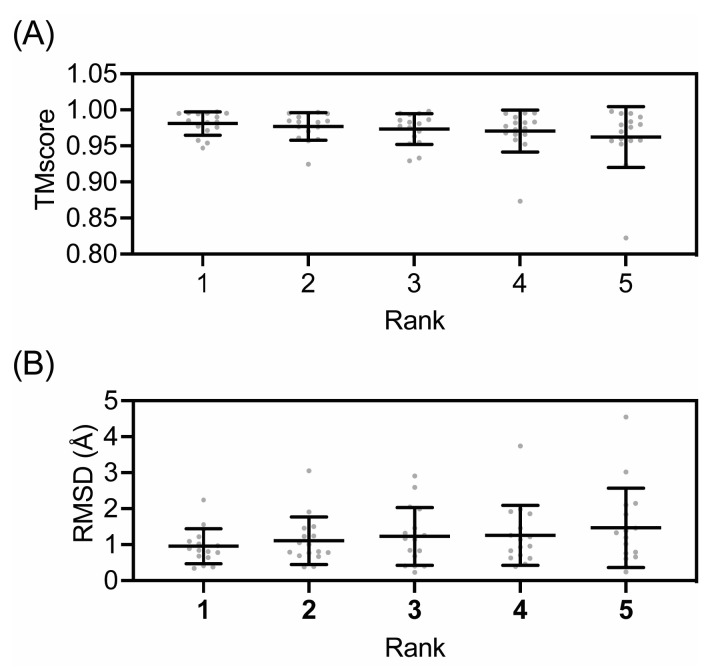
TMscore and RMSD across Non-ATP Grasp Ligase predictive models. TMscores (**A**) and RMSD (**B**) from the 5 models produced by AlphaFold compared to their cognate experimental structures.

**Figure 6 biomolecules-13-01243-f006:**
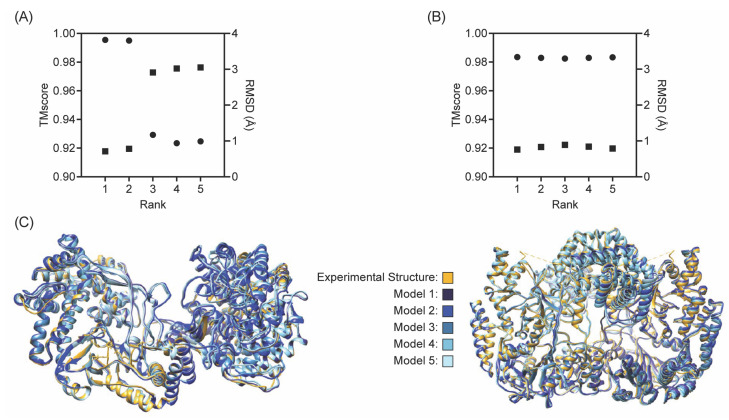
Performance with CylM and NisB. (**A**) US-align results for CylM (PDB ID: 5DZT) and all 5 AlphaFold models (TMscore: circles, RMSD: squares). (**B**) US-align results for NisB (PDB ID: 4WD9) and all 5 AlphaFold models. (**C**) All models aligned to reported experimental structure for both CylM and NisB.

## Data Availability

PDB files used in this study are available at https://www.rcsb.org. Scores for predicted structure versus experimental alignments are available in the Appendix A.

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
