# Peer review of "AlphaFold Accurately Predicts the Structure of Ribosomally Synthesized and Post-Translationally Modified Peptide Biosynthetic Enzymes"

_biomolecules, 2023, doi:10.3390/biom13081243_

Round 1

Reviewer 1 Report

The article is well written, clear and concise. The choice of scientific task - checking the accuracy of someone else's program - is surprising. As a rule, the proof of the quality of this or that program lies on its authors. The scientific value of such work seems to be low. 

Author Response

We appreciate the reviewer's insights regarding our research. It is crucial to understand that protein families vary in their prominence within protein structure research. While some are extensively studied, others, like the RiPP biosynthetic enzymes, remain understudied. Evaluating a machine-learning-based generalized tool like AlphaFold for its efficacy in predicting the structure of such specialized proteins holds significant value. Although the developers of AlphaFold might not have specifically optimized for every unique protein class, our findings confirming its accuracy for this particular subclass offer a meaningful contribution to the community. 

We have expanded the discussion section to highlight the above points. 

Reviewer 2 Report

The manuscript is well written and organized. This reviewer would have liked a bit more introduction to this fascinating topic. For instance, the precursor peptide may have both a leader and follower peptide. The leader peptide is recognized and bound by modifying enzymes and may be responsible for keeping multiple modifications occuring in the proper order. RiPP and associated enyzmes are receiving a good deal of attention so that the use of AlphaFold to predict structures of these enymes is an important contribution.

It is somewhat surprising that addition of AMBER to relax predicted structure does not improve the resulting TM and RMSD. One thing that was interesting in Figure 3 was that the distribution of the accuracy of models appear multimodal with clustering into either ‘good’ or ‘bad’ models. It would be nice to have some comment on why but not necessary to the results.

A few very minor comments:

line 150.  Add something like “’…more so than monomers from experimental…”
In  figures S1-S6, the captions should have US-align rather than USalign.

Author Response

We thank the reviewer for the feedback on our manuscript. We have added more discussion in the introduction about the function of RiPP biosynthetic enzymes. We also thank the reviewer for pointing out the apparent multimodal distribution of the comparisons between models and experimental data. This distribution is due to some monomers in the experimental data adopting different conformations. The models tend to align well with one conformation and less well with others, therefore we see clusters of similarity around how similar the experimental monomers from the same complex compare with each other. We have added this discussion to the text. We have also corrected our typographical errors. 

Reviewer 3 Report

The manuscript submitted by C. H. Gordon, E. Hendrix, Y. He and M. C. Walker describes how AlphaFold can be used to predict structures of enzymes related to PTMs introduction in RiPPs.

There is an interesting discussion, in the Introduction, of the emerging literature devoted to AlphaFold benchmarking – how reliable are the predictions.

The computational work described in the manuscript is quite impressive. The Authors used ColabFold in many different ways, with templates and without templates, with energy minimization of without energy minimization, with 3 to 72 recycles. The Authors handled both monomeric and dimeric proteins.

The results of all this work are, unfortunately, quite modest. In fact, the variations of TMscores and Rmsds are extremely modest. At this regard, it would be necessary to provide some statistical estimates about the significance of the variations.

The key finding of this study is that the predictions remain consistent even when the inputs to AlphaFold and ColabFold are altered. This is interesting and deserves to be published.

I strongly suggest to stress on this point in a revised version of the manuscript.

Author Response

We thank the reviewer for the comments on our manuscript. We agree that the improvements we see when changing the parameters are modest, and in fact are not statistically significant. We feel that the confirmation of the utility of ColabFold for predicting the structures of RiPP biosynthetic enzymes is a useful contribution to the field. We have updated the text of our manuscript to reflect that even with the default parameters, accurate predictions are made.